# Genetic Analysis Reveals Differences in CD8^+^ T Cell Epitope Regions That May Impact Cross-Reactivity of Vaccine-Induced T Cells against Wild-Type Mumps Viruses

**DOI:** 10.3390/vaccines9070699

**Published:** 2021-06-25

**Authors:** Patricia Kaaijk, Maarten E. Emmelot, Jeroen Kerkhof, Cécile A.C.M. van Els, Hugo D. Meiring, Jelle de Wit, Rogier Bodewes

**Affiliations:** 1Centre for Infectious Disease Control, National Institute for Public Health and the Environment, 3721 MA Bilthoven, The Netherlands; maarten.emmelot@rivm.nl (M.E.E.); jeroen.kerkhof@rivm.nl (J.K.); cecile.van.els@rivm.nl (C.A.C.M.v.E.); jelle.de.wit@rivm.nl (J.d.W.); rogier.bodewes@rivm.nl (R.B.); 2Faculty of Veterinary Medicine, Utrecht University, 3584 CL Utrecht, The Netherlands; 3Intravacc (Institute for Translational Vaccinology), 3721 MA Bilthoven, The Netherlands; hugo.meiring@intravacc.nl

**Keywords:** T cell epitopes, T cell response, cytotoxic CD8^+^ T cells, mumps virus, Jeryl–Lynn mumps vaccine, wild-type strains, HLA-I, immunoinformatics, antigenic variation, T cell cross-reactivity

## Abstract

Nowadays, mumps is re-emerging in highly vaccinated populations. Waning of vaccine-induced immunity plays a role, but antigenic differences between vaccine and mumps outbreak strains could also contribute to reduced vaccine effectiveness. CD8^+^ T cells play a critical role in immunity to viruses. However, limited data are available about sequence variability in CD8^+^ T cell epitope regions of mumps virus (MuV) proteins. Recently, the first set of naturally presented human leukocyte antigen Class I (HLA-I) epitopes of MuV was identified by us. In the present study, sequences of 40 CD8^+^ T cell epitope candidates, including previously and newly identified, obtained from Jeryl–Lynn mumps vaccine strains were compared with genomes from 462 circulating MuV strains. In 31 epitope candidates (78%) amino acid differences were detected, and in 17 (43%) of the epitope candidates the corresponding sequences in wild-type strains had reduced predicted HLA-I-binding compared to the vaccine strains. These findings suggest that vaccinated persons may have reduced T cell immunity to circulating mumps viruses due to antigenic differences.

## 1. Introduction

Since the introduction of the measles, mumps and rubella (MMR) vaccine in national immunization programs, mumps incidence has dramatically decreased. However, in the last decade, again several mumps outbreaks have been reported, especially among vaccinated young adults. Waning of vaccine-induced immunity is considered to play a central role in the re-emergence of mumps among vaccinated young adults [1,2,3,4,5]. However, antigenic differences between the currently circulating wild-type viruses and the vaccine virus may further lead to reduced vaccine-induced immunity. Several studies indicate that antigenic differences may impact the antibody response; the level of antibody neutralization of outbreak virus strains was found to be lower than for the vaccine strain [6,7,8,9]. Vaccine-induced immunity to mumps may also be affected by differences in antigenic peptides recognized by T cells (T cell epitopes) between vaccine and circulating mumps virus (MuV) strains [10]. For instance, sequence variations in CD8^+^ T cell epitopes of influenza A virus and human immunodeficiency virus 1 (HIV-1) have been demonstrated to be associated with escape from virus-specific CTLs by abrogation of T cell receptor (TCR) recognition of the epitopes [11,12]. This indicates that viruses can escape CD8^+^ T cell-mediated immunity.

Several studies have shown the presence of T cells reactive against MuV after natural infection and vaccination [13,14]. CD8^+^ T cells play a critical role in mediating viral clearance following many respiratory virus infections [15]. Previously, we showed that strong polyfunctional CD8^+^ T cells dominate the T cell response in mumps patients, whereas in recently vaccinated children the (polyfunctionality of the) CD8^+^ T cell response was considerably lower [13]. It might be that the induction of a mumps-specific CD8^+^ T cell response is suboptimal after vaccination, and that the effectiveness of this response might be further hampered by genetic differences between vaccine strains and currently circulating MuV strains. Although MuV is considered an antigenic stable monotype virus, antigenic differences leading to decreased neutralization capacity of vaccine-induced antibodies have been found [6,7,8,9]. It is unknown whether amino acid variations occur in T cell epitope regions of wild-type mumps viruses that may lead to abolished TCR recognition by vaccine-induced CD8^+^ T cells.

Analysis of T cell epitopes can be a useful tool to characterize the CD8^+^ T cell response after vaccination, and explore the cross-reactive potency of the T cells induced by vaccination towards circulating strains. Recently, we described the first set of T cell epitopes of MuV [16,17]. CD8^+^ T cell epitopes of MuV were identified by eluting peptides from HLA Class I (HLA-I) molecules from antigen-presenting cells that were infected with MuV, using a genotype G outbreak strain. A total of 41 MuV peptides that were naturally processed could be confirmed by comparison of the mass spectra of the corresponding synthetic peptides [17]. In parallel, we performed a similar experiment to identify vaccine-derived MuV epitope candidates obtained by elution from HLA-I molecules from cells infected with Jeryl–Lynn (JL) vaccine MuV. The JL mumps vaccine consists of two viral components, i.e., the minor JL2 and major JL5 vaccine MuV strains (both Genotype A), which differ by over 400 nucleotides [18]. In the present study, we compared amino acid sequences of the set of T-cell epitope candidates derived from JL mumps vaccine virus strains with corresponding sequences of globally circulating MuV strains.

Sequence differences in CD8^+^ T cell epitopes between vaccine and wild-type MuV strains may lead to diminished recognition of wild-type mumps MuV by T cells that were induced by vaccination. Cytotoxic CD8^+^ T cells recognize epitopes presented by HLA-I molecules on the surface of cells. HLA molecules display extensive polymorphism in domains responsible for antigen binding and interactions with variable regions of the T cell receptors. The selection of epitopes that is presented markedly varies across the different HLA alleles. However, several HLA-I supertypes have been described that represent a group of HLA molecules which bind to a similar set of peptides. Immunoinformatic tools can predict the binding affinity of peptides to the various HLA molecules. In recent years, the accuracy of these in silico prediction methods has increased strikingly [19]. In the present study, we used the NetMHC-pan-4.1 tool to predict the binding of the identified T-cell epitope candidates derived from the JL vaccine MuV. Epitope binding was predicted to all HLA-I (A and B) supertypes, i.e., HLA-A*01:01, HLA-A*02:01, HLA-A*03:01, HLA-A*24:02, HLA-A*26:01, HLA-B*07:02, HLA-B*08:01, HLA-B*27:05, HLA-B*39:01, HLA-B*40:01, HLA-B*58:01, HLA-B*15:01 as well as to the two HLA-C alleles, HLA-C*03:04, or HLA-C*07:02 that were expressed on the antigen-presenting cells used to identify the MuV epitope candidates. In addition, we explored whether observed differences in amino acid sequences within the potential T cell epitopes among various wild-type MuV strains led to differences in predicted HLA-I binding.

## 2. Materials and Methods

### 2.1. Set of CD8^+^ T Cell Epitope Candidates of Mumps Virus (MuV) of Jeryl–Lynn (JL) Vaccine

#### 2.1.1. Generation of Mumps Virus-Infected B-Lymphoblastoid Cell Line

JL-MuV virus stock was generated by harvesting supernatant of MuV-inoculated Vero cells as described previously [20]. JL-MuV virus was harvested at peak cytopathic effect, centrifuged (485× *g*), and filtered (5 μm). Virus stocks were aliquoted and stored at −80 °C until use. Epstein–Barr virus-transformed B-lymphoblastoid cell line (BLCL), as efficient antigen-presenting cells, was generated as described before from PBMCs of a healthy donor with the following HLA-typing: HLA-A*01:01, A*02:01, B*07:02, B*40:01, C*03:04, C*07:02 [17]. Approximately 1.3 × 10^9^ BLCL cells were infected with JL-MuV at a multiplicity of infection of 2 for 72 h in Roswell Park Memorial Institute (RPMI) 1640 medium supplemented with 100 U/mL penicillin, 100 μg/mL streptomycin, and 292 μg/mL L-glutamine (Thermo Fisher Scientific, Waltham, MA USA and 2% FCS (HyClone; Cytiva, Freiburg, Germany). This procedure yielded 31% MuV-infected cells, based on expression of MuV-F protein measured by flow cytometry as previously described [20]. Infected BLCL cells were harvested, washed 3 times in cold phosphate-buffered saline, pelleted, snap-frozen, and stored at −70 °C. Cell viability was high, and MuV infection did not affect the expression of total HLA Class I molecules.

#### 2.1.2. Isolation of Human Leukocyte Antigen Class I-Bound Peptides

Human leukocyte antigen Class I-peptide complexes (p:HLA-I) were isolated from MuV-infected BLCL cells using purified antihuman HLA-A/B/C monoclonal antibody (clone W6/32) as described previously [21]. In brief, cells were solubilized in lysis buffer containing NP-40 and protease inhibitors (Thermo Fisher Scientific, Waltham, MA, USA). After centrifugation at 10,000× *g* for 1 h at 4 °C, supernatants were precleared with cyanogen bromide-activated and Tris-blocked (control) sepharose beads (GE Healthcare, Freiburg, Germany), and beads coupled to normal mouse serum (Jackson Immunoresearch Laboratories, PA, USA), respectively, and cleared with beads coupled to W6/32. Immunoprecipitated p:HLA-I were eluted from the beads with 10% acetic acid, and peptides were subsequently collected by passage over a 10-kDa molecular weight cutoff membrane filter. The filtrated HLA-I peptide eluates were concentrated by vacuum centrifugation.

#### 2.1.3. Two-Dimensional Reversed-Phase Liquid Chromatography-Mass Spectrometry and Assignment to Mumps Virus Protein Sequences

Subsequently, the eluted peptide pool was fractionated into 13 fractions by high pH Reversed Phase chromatography with each fraction being subjected to low pH Reversed Phase nanoscale LC-MS analysis for peptide identification. MS spectra were acquired on an Orbitrap Fusion Lumos mass spectrometer (ThermoScientific, Waltham, MA, USA) with an Orbitrap readout for the MS scans. Multiply charged species (2^+^–5^+^) with an intensity >5000 counts were selected for collision-induced dissociation, with an ion trap readout for the MS2 scan. MS data were submitted to the Protein Discoverer 2.1 application (ThermoScientific, Waltham, MA USA) for peptide identification utilizing databases containing proteins from Homo sapiens (TaxID = 9606) and the MuV strains, Jeryl–Lynn (TaxID = 11,168). Precursor and fragment mass tolerances were 5 ppm and 0.4 Da, respectively. Dynamic modifications were set for methionine (oxidation) and asparagine (deamidation) and pyro-Q for the peptide N-termini.

Deamidation of asparagine (N) into aspartic acid (D) is a post-translational modification that can occur either spontaneously or through enzymatic reaction in proteomics workflows, but also during the MHC Class I processing route of cells during back-transport from the endoplasmic reticulum (ER) to the cytosol as part of the ER-associated degradation pathway. The modified proteins are then processed by the proteasome, yielding deamidated HLA ligands [22,23].

The eluted peptides that could be assigned to MuV proteins were selected for final confirmation by MS analysis using synthetic analogues (JPT, Berlin, Germany) for comparison. Only the eluted candidate MuV peptides that had a similar *m/z* ratio, retention time, and mass spectrum as the corresponding synthetic MuV peptides were confirmed as HLA Class I presented-MuV epitope.

#### 2.1.4. Previous Described Set of Mumps Virus CD8^+^ T Cell Epitope Candidates

In addition, a set of 41 peptides which was eluted from HLA-I molecules of antigen-presenting cells infected with a mumps genotype G virus detected during an outbreak in the Netherlands outbreak MuV strain (MuVi/Utrecht.NLD/40.10) was previously described [17]. From this set of 41 peptides, the peptides were selected that were, with exactly the same amino acid sequence, also present in (one of) the JL2/JL5 vaccine MuV strain(s). The same antigen-presenting cell line was used for both HLA-I elution experiments and had the following HLA-typing: HLA-A*01:01, A*02:01, B*07:02, B*40:01, C*03:04, C*07:02 [17]. By these two approaches a set of CD8^+^ T cell epitope candidates of JL vaccine MuV was identified.

### 2.2. Comparison of Amino Acid Sequences of CD8^+^ T Cell Epitope Candidates with Various Wild-Type MuV Strains

Near complete MuV genomes from 539 mumps viruses (minimum sequence length 14,000 nucleotides) available on GenBank were downloaded on 31 December 2020. In addition, two near complete genomes from mumps viruses detected in the Netherlands were added to the dataset (MuVi/Sint Philipsland.NLD/02.08 and MuVi/Utrecht.NLD/40.10 (GenBank accession IDs respectively MW819866 and MW261742), resulting in total 541 near complete mumps virus genomes. Only mumps viruses from which the genome sequence was determined directly from clinical materials or after a low number of passages were included in the analysis, resulting in in total 462 mumps viruses that were analyzed (Appendix A). Mumps virus (strain Jeryl–Lynn) live vaccine minor component JL (GenBank accession 345,290) and mumps virus (strain Jeryl–Lynn) live vaccine major component (GenBank accession AF338106) were used as representative for the mumps virus vaccine strains JL2 and JL5 respectively [24]. Sequences were aligned using the MAFFT (multiple alignment using fast Fourier transform) online service [25]. Aligned nucleotide sequences of MuV genomes were subsequently translated to amino acid sequences using MEGA7 for each gene separately [26]. Variation between amino acid sequences of MuV CD8^+^ T cell epitope candidates and corresponding deduced amino acid sequences in wild-type MuV strains was analyzed in BioEdit version 7.2.5 [27]. For a limited number of recent MuV strains included in the analysis, sequence data was not available for all epitope candidates. These viruses were still included in the analysis except for epitopes with no complete sequence data.

### 2.3. Prediction of Human Leukocyte Antigen Class-I Binding of T Cell Epitope Candidates of MuV

Identified peptides with sequence variations were tested for predicted binding to the HLA-I (A and B) supertypes, (i.e., HLA-A*01:01, HLA-A*02:01, HLA-A*03:01, HLA-A*24:02, HLA-A*26:01, HLA-B*07:02, HLA-B*08:01, HLA-B*27:05, HLA-B*39:01, HLA-B*40:01, HLA-B*58:01, HLA-B*15:01 as well as to the two HLA-C alleles, HLA-C*03:04, or HLA-C*07:02 that were expressed on the antigen-presenting cells used to identify the MuV epitope candidates) using the NetMHCpan 4.1 server (used on 18 February 2021) [19]. For this analysis, only genomic sequences obtained from Genbank were considered and thus not any (post-translational) modifications. A threshold for binding peptides was set on rank <2%. The % rank scores of a sequence is computed by comparing its prediction score to a distribution of prediction scores for the MHC in question, estimated from predicted binding affinity values from a set of random natural peptides of 125,000 8–12-mer random natural peptides. Peptides with lower rank than 0.5% were considered as strong binder and with rank > 0.5% ≤ 2% were considered to bind weakly to HLA-I [19,28]. Rank scores of the different peptides for the predicted binding to multiple HLA alleles were visualized with a heatmap. This heatmap was created using the heatmap.2 package in R version 4.0.2 (R script available upon request) [29].

## 3. Results

In the present study, a total of 20 MuV peptides, including six novel epitope candidates, were identified with advanced mass spectrometry by elution of peptides from HLA Class I molecules (HLA-I) of JL vaccine MuV-infected cells (Table 1). The other 14 from this set of 20 JL vaccine MuV epitope candidates have already been identified and were included in the previously described set of 41 peptides [17] eluted from HLA-I molecules of antigen-presenting cells infected with a genotype G outbreak MuV strain. From this set of 41 peptides eluted from Genotype G outbreak MuV strain another 20 epitope candidates had identical amino acid sequences as the JL vaccine MuV strain(s). This resulted in a total of 40 CD8^+^ T cell epitope candidates associated with JL2/JL5 vaccine MuV strains (Table 1 and Figure 1). All these identified epitope candidates were obtained by HLA-I ligand elution of antigen-presenting cells with the following HLA-typing: HLA-A*01:01, A*02:01, B*07:02, B*40:01, C*03:04, C*07:02. Therefore, peptides presented by other HLA-I molecules have not been determined experimentally. Next, amino acid sequences of these 40 epitope candidates of JL vaccine MuV strains were compared with corresponding amino acid sequences in a large set of circulating MuV strains to identify sequence variations (Appendix A). In 31 out of 40 epitope candidates (78%), amino acid differences were detected between the JL2/JL5 vaccine MuV strains and wild-type MuV strains (Table 1). 

Subsequently, we explored whether the sequence differences that were observed in the potential T cell epitopes of the vaccine MuV strains in comparison with the corresponding sequence variants found in the circulating strains led to reduced predicted binding to (one or more) HLA-I molecule(s), and may therefore be less likely to induce an adequate T cell response. In 17 out of the 40 (43%) JL vaccine MuV epitope candidates, corresponding regions in wild-type strains showed to have a reduced predicted HLA-I-binding compared to sequences in JL2/JL5 vaccine MuV strains (Table 1, Figure 2). Two length variants of KPRTSTPVTEF (i.e., KPRTSTPVT; RTSTPVTEF) are represented in this set of 17 JL vaccine MuV epitopes. Additionally, amino acid sequences of 3 out of these 17 epitopes differ between the two (JL2/JL5) vaccine component variant strains. 

Interestingly, sequence differences were found in all nine T cell epitope candidates of non-structural V/phospo (P)/I (V/P/I) proteins, which in 8/9 cases (89%) also led to reduced predictive HLA-I binding of the corresponding peptides in circulating strains compared to the epitope candidates of the JL2/JL5 vaccine MuV strains. For instance, various wild-type sequence variants of JL vaccine MuV epitope V/P/I27-35 (TPIQGTNSL) showed to have reduced predicted binding to HLA-B*07:02, HLA-B*39:01 and HLA-C*03:04 molecules. Amino acid variation in four out of five (80%) epitope candidates of hemagglutinin-neuraminidase (HN) protein also led to reduced predicted HLA-I-binding of the variants in circulating strains. In contrast, none of the observed circulating wild-type sequence variants of the JL2/JL5 vaccine MuV epitope candidates of the fusion (F) and matrix (M) proteins led to reduced predicted binding to HLA-I. In addition, in only 4 of the 10 epitope candidates of the large (L) protein, variation sequences found in wild-type strains did lead to reduced predicted binding to HLA-I. One of the two epitope candidates of nucleoprotein (NP) led to reduced predicted HLA-I-binding (Figure 1B and Figure 2, Table 1).

There were remarkably few sequence variations observed in the epitope candidates that led to reduced binding to the most common allele, HLA-A*0201, while relatively a large number of sequence variants of the epitopes were found leading to reduced binding to HLA-B*07:02, also a common HLA-I molecule (Figure 2).

## 4. Discussion

In the present study, 40 CD8^+^ T cell epitope candidates derived from JL2/JL5 vaccine MuV strains were characterized and amino acid sequences of these epitope candidates were compared with that of circulating MuV strains. Furthermore, the impact of the sequence differences observed in the circulating strains on predicted HLA-I binding affinity was determined. Binding between HLA and antigenic peptides is the most selective step in the antigen presentation pathway, and prediction of peptide binding to HLA is a powerful utility to predict the possible T cell immunogenicity of peptides [19]. Reduced predicted HLA-I binding affinity of protein sequences within the regions of naturally processed candidate CD8^+^ T cell epitopes of circulating MuV strains compared to mumps vaccine strains may therefore be indicative for impaired CD8^+^ T cell immunogenicity. Amino acid differences were found in 78% (31/40) of the JL vaccine MuV epitope candidates when the sequences were compared with a total of 462 MuV genomes that were found available on GenBank. Moreover, 43% percent (17/40) of the JL vaccine MuV epitopes candidates showed to have sequence differences in circulating MuV strains that led to (varying degree of) reduction in predicted HLA-I-binding as compared to the vaccine MuV strains. In line with this, in another study by our group, we showed a relative high non-synonymous mutation rate in regions encoding HLA Class I epitopes of measles virus [30]. This may indicate molecular evolution of these viruses under pressure of cellular immunity.

Non-synonymous mutations in viral genomes create molecular diversity in viral proteins enabling viruses to adapt to the host’s immune response. Mumps viruses have a relatively low nucleotide substitution rate compared to other RNA viruses, but also in mumps viruses mutations do occur [31]. These mutations, which can result in antigenic altered viruses that can replicate or even cause disease in previously immune hosts. Perhaps this is more likely to occur if a person has acquired immunity exclusively through vaccination with a vaccine strain that differs antigenically from the currently circulating virus strains. A study in guinea pigs showed that antibodies to the JL vaccine effectively neutralized virus from a Genotype G outbreak strain and other recent wild-type strains of different genotypes, but reduced neutralizing titers were observed compared to the vaccine strain indicating an antigenic divergence from the vaccine strains [9]. The already waning vaccine-induced immunity in young adults due to time (≥10 years) elapsed after their last MMR dose [1,2] together with the reduced immunity against antigenic different circulating MuV strains can lead to an even higher susceptibility to MuV infection in this age group.

To our knowledge, only one study investigated the possible consequences of antigenic differences in MuV strains on T cell immunity by applying immunoinformatics techniques [10]. Homan et al. (2014) [10] evaluated differences in HLA binding affinity and cathepsin cleavage in the HN protein of JL2/JL5 and Rubini vaccine strains versus four wild-type mumps isolates. They found distinct differences between the predicted HLA-I and HLA-II binding patterns in the HN protein of various MuV strains. Based on their findings, they considered JL5 vaccine strain as an outlier, with immunogenic features arising from a small number of amino acid changes that distinguish it from other MuV strains. They paid special attention to the substitution of I279T and I287V between JL5 HN and wildtype HN that may lead to a mismatch of CD8^+^ responses, but in our study no CD8^+^ T cell epitope candidates were identified located in these regions of the HN protein. In our study, five HN epitope candidates derived from JL vaccine strains were eluted from HLA-I molecules, from which four of these peptides showed to have sequence differences with wild-type MuV strains and these four wild-type variants all led to reduced HLA-I binding compared to the JL vaccine MuV epitope candidates. A comparison of the other MuV proteins showed that especially JL vaccine MuV candidate epitopes of the V/phospo (P)/I proteins varied among JL vaccine and wild-type strains and often (89%) led to reduced predictive HLA-I binding of the corresponding peptides in circulating strains. In contrast, none of the found wild-type sequence variants of the JL vaccine MuV epitope candidates of the F and M proteins led to reduced predicted binding to HLA-I. In addition, in only four of the ten epitope candidates of the L protein and one of the two epitope candidates of N protein, variation sequences found in wild-type strains led to reduced predicted binding to HLA-I. Thus, in our study, especially JL vaccine MuV epitope candidates of the V/P/I and HN MuV proteins showed to have sequence differences with wild-type MuV strains that led to reduced predictive HLA-I binding. However, a more detailed in vitro characterization of the impact of the observed sequence variations on CD8^+^ T cell recognition of virally-infected cells, and subsequent T cell activation is needed.

So far, we confirmed immunogenicity of 11 MuV T cell epitope candidates by showing the induction of a strong T-cell response and thus recognition by a cognate T cell receptor (TCR) [17,32] (Table 1). Eight of these eleven (73%) confirmed T cell epitopes showed sequence difference(s) between the vaccine strain(s) and one or more circulating strain(s). It should be realized that vaccine-induced T cells reactive against a vaccine epitope might not recognize the corresponding sequence variant of a wild-type strain, even though the wild-type sequence variant has a better predicted HLA-I binding affinity than the vaccine epitope. The TCR of the MuV vaccine-specific T cells may simply not recognize the wild-type sequence variants with mutation(s) anymore due to loss of contact residues. On the other hand, a certain flexibility of the TCR can occur as well that allows recognition of epitope variants with some mutations as has been described for HIV-specific CD8^+^ T cells [33]. 

Deamidation of asparagine (N) into asparagine acid (D) is a predominant known post-translational modification that can occur either as a spontaneous non-enzymatic reaction (e.g., in most in vitro and in vivo experiments) or as part of an enzyme-controlled biological process (e.g., deglycosylation of N). In the present genetic analysis, only genomic sequences obtained from Genbank were considered to evaluate sequence variability in T cell epitope regions of MuV and thus not any (post-translational) modifications. For this purpose, all available MuV sequences with genomes >14,000 nucleotides were downloaded from GenBank, but the number of available sequences from mumps viruses was low compared to many other (viral) pathogens. From multiple outbreaks, none or only one (near) complete MuV genome was available while for a few outbreaks a relatively high number of sequences was available. Therefore, it is complex to draw conclusions about the frequency of certain epitope variants in wild-type virus strains based on this dataset. However, results of this study clearly indicate that molecular surveillance can be also useful to study the possible impact of genetic variation on the cellular immune response to mumps viruses.

Summarizing, this genetic analysis shows that there are differences in T cell candidates epitopes of JL2/5 vaccine MuV strains and corresponding sequences in wild-type virus strains that may contribute to reduced immunity of vaccinated persons to antigenic different circulating MuV strains. Nevertheless, the JL MuV vaccine has proven itself to be highly efficacious resulting in a 99% decline in disease incidence compared to the pre-vaccine era [34]. In line with this, studies indicate that a third dose of the MMR vaccine may help to prevent and/or controlling mumps outbreaks [2] by improving immunity to MuV [3,35,36]. However, waning vaccine-induced immunity caused by reduced immunity to antigenic different circulating MuV strains may accelerate loss of vaccine effectiveness and should be monitored. To improve immunity to mumps to prevent and control mumps outbreaks in future, the development of a polyvalent MuV vaccine combining diverse genotypes and currently circulating strains in circulation has been suggested as a solution [37,38].

## Figures and Tables

**Figure 1 vaccines-09-00699-f001:**
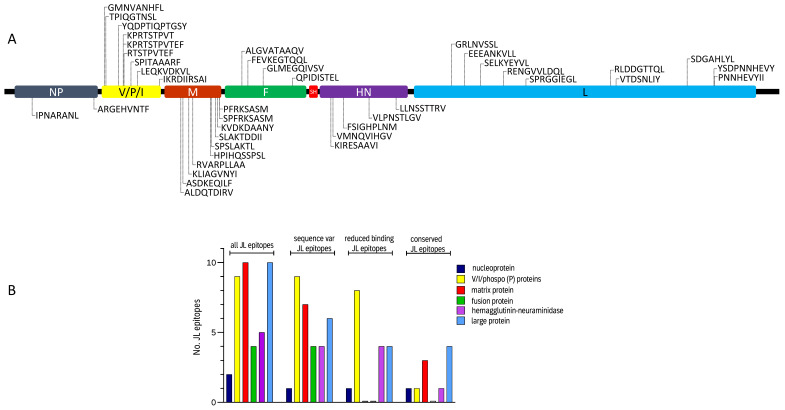
Characteristics of identified CD8^+^ T cell epitope candidates of Jeryl–Lynn (JL) vaccine mumps virus (MuV). Location of the JL vaccine MuV epitope candidates within the MuV proteins (**A**). Distribution of JL vaccine MuV candidate epitopes among the various MuV proteins. No epitope candidates of small hydrophobic (SH) protein (57 amino acids long) were identified (**B**). Number of total JL vaccine MuV candidate epitopes among the various MuV proteins are presented, as well as number of JL vaccine MuV candidate epitopes that show to have sequence differences in circulating MuV strains, number of JL vaccine MuV candidate epitopes that show to have sequence differences in circulating MuV strains that led to reduced predicted binding to one or more human leukocyte antigen Class I molecules, and number of JL vaccine MuV candidate epitopes with conserved amino acid sequences, i.e., epitopes that show no sequence differences in any of the circulating MuV strains. NP, nucleoprotein; V/P/I, V/phospo (P)/I proteins; M, matrix protein; F, fusion protein; SH, small hydrophobic protein; HN, hemagglutinin-neuraminidase; L, large protein.

**Figure 2 vaccines-09-00699-f002:**
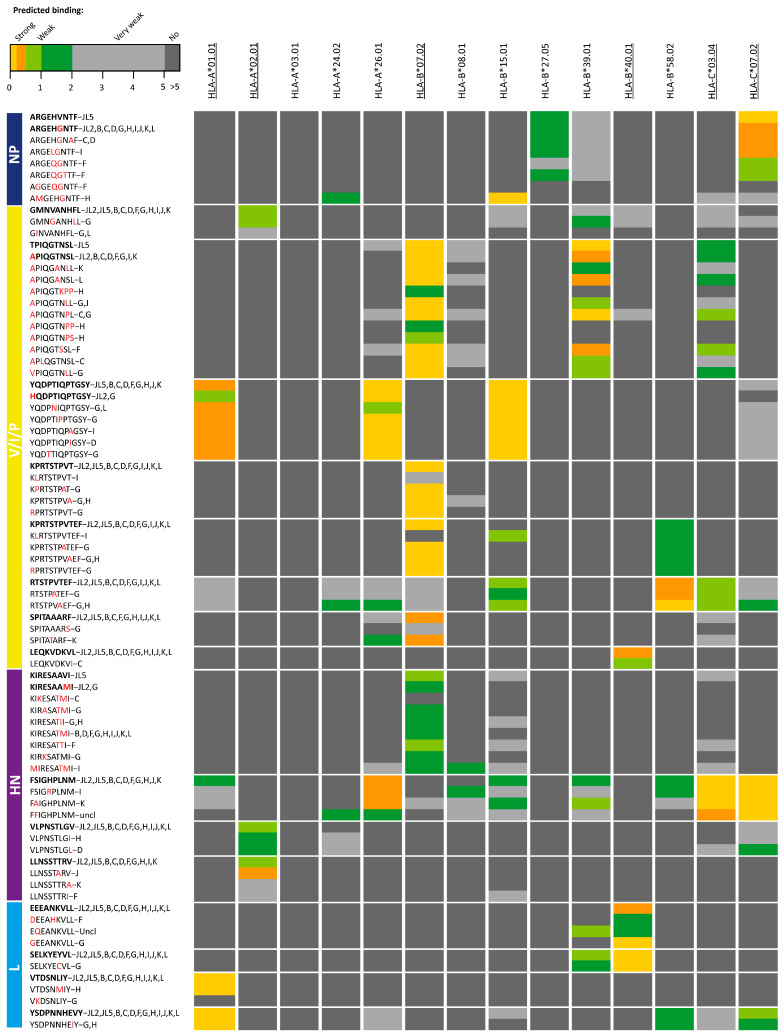
Heat map of epitope candidates of the JL2/JL5 mumps vaccine viruses in which amino acid sequence variants were found in circulating strains that led to reduced predicted binding to human leukocyte antigen Class I (HLA-I) molecule(s). Color scale indicates differences in predicted HLA-I binding peptides with lower rank (yellow) representing strong predicted HLA-I binding and dark grey no binding (>5% rank). Bars on the left indicate proteins where epitope candidates are part of. Epitope variants detected in Jeryl Lynn (JL) mumps vaccine strains are indicated as the first variant of each epitope and in bold. Behind each epitope variant is also indicated in which genotype(s) epitope variants were detected (uncl = unclear; Mumps rubulavirus—UniProt KF878082). Differences in amino acid sequences compared to epitope candidates of the JL2/JL5 mumps vaccine viruses are indicated with red character. HLA-I molecules that were expressed by the cell line used to identify the epitope candidates are underlined. NP, nucleoprotein; V/P/I, V/phospo (P)/I proteins; M, matrix protein; F, fusion protein; SH, small hydrophobic protein; HN, hemagglutinin-neuraminidase; L, large protein.

**Table 1 vaccines-09-00699-t001:** Sequence variation and reduced MHC-I binding of the 40 CD8^+^ T cell epitope candidates of JL MuV.

MuV Protein	Amino Acid Sequence	X-Mer	Protein Location	Described [Ref]/Novel	JL Eluted	Seq Variation	Reduced Binding
N	**IPNARANL**	8	115–122	[17]	no	no	no
N	ARGEHVNTF	9	524–532	novel	yes	yes	yes
V/P/I	**GMNVANHFL**	9	17–25	[17]	no	yes	yes
V/P/I	**TPIQGTNSL**	9	27–35	novel	yes	yes	yes
V/P/I	YQDPTIQPTGSY	12	111–122	[17]	no	yes	yes
V/P/I	KPRTSTPVT	9	142–150	[17]	yes	yes	yes
V/P/I	**KPRTSTPVTEF**	11	142–152	[17]	yes	yes	yes
V/P/I	RTSTPVTEF	9	144–152	[17]	no	yes	yes
V/P/I	SPITAAARF	9	194–202	[17]	no	yes	yes
P	LEQKVDKVL	9	235–243	novel	yes	yes	yes
P	IKRDIIRSAI	10	382–391	novel	yes	yes	no
M	**ALDQTDIRV**	9	108–116	[17]	yes	yes	no
M	ASDKEQILF	9	120–128	[17]	no	no	no
M	**KLIAGVNYI**	9	161–169	[17]	no	yes	no
M	RVARPLLAA	9	187–195	[17]	yes	no	no
M	HPIHQSSPSL	10	303–312	[17]	yes	no	no
M	SPSLAKTL	8	309–316	[17]	yes	yes	no
M	SLAKTDDII	9	337–345	[17]	no	yes	no
M	KVDKDAANY	9	350–358	[17]	no	yes	no
M	SPFRKSASM	9	364–372	[17]	yes	yes	no
M	PFRKSASM	8	365–372	[17]	no	yes	no
F	ALGVATAAQV	10	112–121	[17]	yes	yes	no
F	FEVKEGTQQL	10	152–161	[17]	no	yes	no
F	**GLMEGQIVSV**	10	253–262	[17]	yes	yes	no
F	QPIDISTEL	9	445–453	[17]	no	yes	no
HN	KIRESAAVI	9	74–82	novel	yes	yes	yes
HN	**VMNQVIHGV**	9	88–96	[17]	yes	no	no
HN	FSIGHPLNM	9	157–165	[17]	no	yes	yes
HN	VLPNSTLGV ^1^	9	326–334	[17]	no	yes	yes
HN	**LLNSSTTRV ^1^**	9	505–513	[17]	no	yes	yes
L	GRLNVSSL ^1^	8	249–256	[17]	yes	no	no
L	EEEANKVLL	9	336–344	[17]	no	yes	yes
L	SELKYEYVL	9	441–449	[17]	no	yes	yes
L	RENGVVLDQL	10	591–600	[17]	no	no	no
L	SPRGGIEGL	9	740–748	[17]	yes	no	no
L	RLDDGTTQL	9	1299–1307	[17]	no	no	no
L	**VTDSNLIY**	8	1336–1343	[17]	no	yes	yes
L	SDGAHLYL	8	1806–1813	[17]	yes	yes	no
L	**YSDPNNHEVY ^1^**	10	1983–1992	[17]	yes	yes	yes
L	PNNHEVYII ^1^	9	1986–1994	novel	yes	yes	no

Sequences in bold indicate MuV T cell epitopes for which immunogenicity have previously been confirmed; APIQGTNLL (and not TPIQGTNSL), and LLDSSTTRV (and not LLNSSTTRV) were proven T cell epitopes. ^1^ For this genetic analysis, only genomic sequences obtained from Genbank were considered and thus not any (post-translational) modifications. Therefore, VLPNSTLGV, and not the eluted VLPDSTLGV variant was considered in this analysis, LLNSSTTRV and not LLDSSTTRV, GRLNVSSL and not GRLDVSSL, YSDPNNHEVY and not YSDPNDHEVY/SDPDNHEVY, and PNNHEVYII and not PDDHEVYII, were considered in the present genetic analysis, because the variants with asparagine acid (D) were not found in the sequences of JL2/JL5 vaccine MuV strains according to Genbank. Note: Mei et al. [22] showed a significant prevalence of the N-linked glycosylation motif N-X-[S/T] (with N as deamidated asparagine and X as any amino acid, except proline) in deamidated HLA ligands originating from formerly glycosylated proteins located in the cell membrane or endoplasmic reticulum. This could function as a N-glycosylation surveillance strategy of the immune system for the host cell proteome. Jeryl–Lynn vaccine (JL); mumps virus (MuV); X-mer, number of amino acids in the T cell epitope candidate; Ref, reference; Seq, sequence; N, nucleoprotein; V/P/I, V/phospo (P)/I proteins; M, matrix protein; F, fusion protein; SH, small hydrophobic protein; HN, hemagglutinin-neuraminidase; L, large protein.

## Data Availability

Mumps virus epitope candidates previously identified by us have been included in the Immune Epitope Database (IEDB), a freely available resource. http://www.iedb.org/home_v3.php (accessed on 20 June 2021)).

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
