# Peer review of "Genetic Analysis Reveals Differences in CD8^+^ T Cell Epitope Regions That May Impact Cross-Reactivity of Vaccine-Induced T Cells against Wild-Type Mumps Viruses"

_vaccines, 2021, doi:10.3390/vaccines9070699_

Round 1

Reviewer 1 Report

Here Kaaijk et al explore the interesting question of whether changes between vaccine strain sequence and circulating mumps virus sequences could lead to differences in the peptides presented by HLA-I to initiate CD8+ T cell responses. Potential epitopes are derived from a new HLA-I peptide elution data set of a vaccine strain infected cell line, as well as previous data derived from the same cell line infected with a circulating strain (produced by the same group). Variations in these epitopes between strains were then found through a translated multiple sequence alignment of Mumps Virus genomes, and their impact on binding affinity to representative HLA-I predicted. It is an important question, however the paper is currently let down by a lack of adequate description of the new LC-MS/MS data set, and some missing explanations detailed below.

  1. In Section 2.1, Some methodologies are currently missing. Although the previous study using a circulating strain of MuV is cited, at least brief methods for the infection, peptide isolation, mass spectrometry and bioinformatic criteria should be included. There is currently no mention of what cell line has been used, only the HLA typing. Whilst this can be found by looking at reference 16, it really should be stated within current manuscript. Also in line 100-101, there is the following statement:
    • “identification utilizing protein databases containing protein sequences from 100 the MuV strains, Jeryl Lynn (TaxID = 11168) were used”
    • Was the human proteome database also included? What modifications were considered? This is especially salient to the note regarding Asn deamidation below table 1 which currently has no context.

  1. In line 145-146, it is stated that 6 novel peptides were identified in the HLA elution data. What is the total number of MuV peptides identified in this experiment? How many of the previously described peptides (i.e. the 34 conserved between genotype G outbreak 148 MuV strain and the JL 149 vaccine MuV strain(s)) were also identified in the HLA elution from the JL 149 vaccine MuV strain infected cells? How many human peptides? A more complete description of the new elution data generated in this study is needed.
  2. For the 7 genotype G outbreak 148 MuV strain peptides that are not shared with the JL 149 vaccine MuV strain(s), do the aligned peptides in the vaccine strain have higher or lower predicted binding affinity? Did these correspond to any of the novel peptides identified in this study? It might be interesting to note these details.
  3. The supplementary table, sheet 1, is very nice, however there appear to be two extra sheets not described by the legend. Were the 2nd and third sheets intentionally included? If so they require inclusion in the legend.
  4. Figure 1B-D would be better represented as a bar chart with Proteins on the x axis, number of peptides on the y axis and 3 series (series 1. No. peptides with sequence differences in circulating MuV strains that led to reduced predicted binding to 1 or more HLA-I molecules, 2.no. of peptides with sequence differences that didn’t lead to reduced predicted binding, 3. Conserved peptides). In its current form the sense of the total number of peptides is lost making it hard to follow.
  5. In figure 2, it should be made more obvious which HLA are expressed by the cell line from which these peptides were identified, perhaps have these 6 on the left-hand-side, and then the remaining on the right hand side? This will help the reader visually see which are the likely binders in the cell line used. For clarity it would also be useful to have some explanation that for HLA not expressed by the cell line in the study, their binders have not been determined experimentally and likely include many that are not described here.
  6. Table 1: The note under table 1 mentions deamidation of asparagine, however this is not well explained, and this is the first and only mention of deamidation. I assume this discussion is due to the fact that the peptides were identified in the LC-MS/MS data as deamidated peptides. Without further explanation either in the methods or the results (or both) this will be unclear to those without an understanding of proteomics.
  7. Table 1: Several of the peptides described in the note as deamidated show the N-X-S/T motif seen for N-linked glycosylations, the enzymatic removal of which leaves a deamidated N. Indeed, the N of LLNSSTTRV is a reported glycosylation site. It would therefore be worth considering that this deamidation may occur in the biological context.

Minor comments:

  1. Referencing seems to be out of sync by 1 due to two references allocated 1 in the reference list
  2. Line 128, “or a limited”, should this be “For a limited”?
  3. Figure 1A and Figure 2, peptide sequences are very small font and hard to read, even when zoomed in due to low resolution
  4. Table 1: Note the 2 in the reference column for TPIQGTNSL lacks [ ]. Is this a reference?

Reviewer 2 Report

In this manuscript, Kaaijk et al., utilize in silico analyses to assess sequence variability among candidate CD8+ T cell epitopes identified from Jeryl Lynn mumps vaccine strains via mass spectrometry between the vaccine strains and circulating mumps virus strains. In my opinion, the experimental design is well done and the results support the notion that sequence differences observed in the circulating strains may have a negative impact on HLA-I binding affinity, and thus CD8+ T cell responses. However, a more detailed characterization of the impact of sequence changes on donor CD8+ T cell recognition of virally-infected cells, and subsequent T cell activation (cytokine secretion and/or CTL activity), in vitro will be needed to support the authors' hypothesis that antigenic changes in circulating strains is a contributing factor to waning mumps vaccine derived immunity.

Minor points:

  1. line 51: the incorrect reference is cited; should be 12 instead of 13
  2. recommend slight increase in font size for the epitopes listed in Figure 1A as I found them difficult to read in their current form
  3. For Figure 1C and 1D, I found the data representation to be slightly confusing. For example, does the 29% and 47% associated V/I/P refer to percentages calculated from the total number of epitope candidates (n=40) or the number of epitope candidates identified for that protein region (n=9)? It may help with clarity, and be informative, to show the later. The authors mention this for one of the MuV proteins (line 266-267), but it would be informative is see this data for all 6 MuV proteins, especially since a high proportion of variability within a set of defined epitope candidates for a protein region rather than the entire set of epitopes may be indicative of a "hot spot" for antigenic variation due to immune pressure.

Round 2

Reviewer 1 Report

Thank you to the authors for their careful consideration and clarifications.

I only have a couple of minor comments for the authors' consideration (note line numbers refer to the revised pdf):

  1. Section 2.1 in the methods is now expanded upon in detail, are lines 93-96 now redundant?
  2. Line 158-159 "The antigen-presenting cells used for both HLA-I elution experiments had the following HLA-typing: HLA-A*01:01, A*02:01, B*07:02, B*40:01, C*03:04, C*07:02 [17]." - my impression was that it was exactly the same cell line? If so, would it be clearer to state as much, e.g. "The same antigen-presenting cell line was used for both HLA-I elution experiments and had the typing..."
  3. Line 282-285 in the footnote of table 1 - this seems more like a discussion point. Perhaps this would be more appropriate in the discussion at line 363?
  4. Line 363 has several typos
  5. The resolution of Figure 1 is still low and obscures some text labels

Author Response

Point-by-point response to reviewer comments

Thank you for your response, and the opportunity to revise our manuscript, entitiled: “Genetic analysis reveals differences in CD8+ T cell epitope regions that may impact cross-reactivity of vaccine-induced T cells against wild-type mumps viruses”, for re-consideration for publication in MDPI Vaccines. We thank the reviewer for his/her careful review and last minor comments, which helped improve the quality of the manuscript.

Please find our answer to comments in blue.

Reviewer #1

Section 2.1 in the methods is now expanded upon in detail, are lines 93-96 now redundant?

We agree with the reviewer that this text is now redundant and removed the text accordingly.

Line 158-159 "The antigen-presenting cells used for both HLA-I elution experiments had the following HLA-typing: HLA-A*01:01, A*02:01, B*07:02, B*40:01, C*03:04, C*07:02 [17]." - my impression was that it was exactly the same cell line? If so, would it be clearer to state as much, e.g. "The same antigen-presenting cell line was used for both HLA-I elution experiments and had the typing..."

We have changed the text according to reviewer’s suggestion.

Line 282-285 in the footnote of table 1 - this seems more like a discussion point. Perhaps this would be more appropriate in the discussion at line 363?

We have now moved this part of the footnoot text to the Discussion section as proposed by the reviewer.

Line 363 has several typos

We have corrected the typos.

The resolution of Figure 1 is still low and obscures some text labels

We think that the resolution of Figure 1, that we separately submitted as PDF file via the online submission tool, should be sufficient. However we ‘cut and pasted’ it in the final text and probably by doing that we lost the higher resolution. We assume that the publisher can solve this and perform this while keeping the good resolution.